# Colorimetric Sensing of Lactate in Human Sweat Using Polyaniline Nanoparticles-Based Sensor Platform and Colorimeter

**DOI:** 10.3390/bios12040248

**Published:** 2022-04-15

**Authors:** Hyun Jung Kim, Insu Park, Seung Pil Pack, Gyudo Lee, Yoochan Hong

**Affiliations:** 1Department of Medical Device, Korea Institute of Machinery and Materials (KIMM), Daegu 42994, Korea; rlaguswnd99@kimm.re.kr; 2Department of Biotechnology and Bioinformatics, Korea University, Sejong 30019, Korea; spack@korea.ac.kr; 3Holonyak Micro and Nanotechnology Laboratory, University of Illinois at Urbana—Champaign, Urbana, IL 61801, USA; insupark@illinois.edu; 4Biological Clock-Based Anti-Aging Convergence Regional Leading Research Center, Korea University, Sejong 30019, Korea; 5Interdisciplinary Graduate Program for Artificial Intelligence Smart Convergence Technology, Korea University, Sejong 30019, Korea

**Keywords:** polyaniline, lactate, sweat, colorimeter, paper sensor

## Abstract

In emergency medicine, the lactate level is commonly used as an indicator of the severity and response to the treatment of hypoperfusion-related diseases. Clinical lactate measurements generally require 3 h for clinical determination. To improve the current gold standard methods, the development of sensor devices that can reduce detection time while maintaining sensitivity and providing portability is gaining great attention. This study aimed to develop a polyaniline (PAni)-based single-sensor platform for sensing lactate in human sweat using a CIELAB color system-based colorimetric device. To establish a lactate sensing platform, PAni nanoparticles were synthesized and adsorbed on the filter paper surface using solvent shift and dip-coating methods, respectively. PAni is characterized by a chemical change accompanied by a color change according to the surrounding environment. To quantify the color change of PAni, a CIELAB color system-based colorimetric device was fabricated. The color change of PAni was measured according to the chemical state using a combination of a PAni-based filter paper sensor platform and a colorimetric device, based on the lactate concentration in deionized water. Finally, human sweat was spiked with lactate to measure the color change of the PAni-based filter paper sensor platform. Under these conditions, the combination of polyaniline-based sensor platforms and colorimetric systems has a limit of detection (LOD) and limit of quantitation (LOQ) of 1 mM, linearity of 0.9684, and stability of 14%. Tbe confirmed that the color of the substrate changes after about 30 s, and through this, the physical fatigue of the individual can be determined. In conclusion, it was confirmed through this study that a combination of the PAni paper sensor platform and colorimeter can detect clinically meaningful lactate concentration.

## 1. Introduction

The clinical evaluation of disease severity is essential for predicting patient mortality and morbidity [1]. In emergency medicine, lactate is a representative biomarker that shows the disease severity and therapeutic response to hypoperfusion-related diseases such as cardiac arrest, trauma, and sepsis [2]. Like other metabolites, such as glucose, cholesterol, and urate, lactate can be oxidized by enzymes into pyruvate to form hydrogen peroxide; that is, it can be decomposed into reactive oxygen species, which plays an essential role in host defense and redox signaling [3,4,5,6,7,8]. Hence, it is important to detect lactate. This can be performed using electrochemical enzyme sensors that bind lactate oxidase (LOx) to the electrode surface [9,10,11]. An enzyme sensor array that simultaneously measures molecular types, including lactate, in biological samples, has been proposed [12]. In addition, electrochemical-based sensor technology has been developed to increase the detection limit or stability against immobilization of enzymes using nanomaterials [13] or various matrices, respectively; however, recently, non-invasive sensors attached to the skin have been spotlighted [14,15].

Furthermore, polyaniline (PAni) has wide applications, such as electronic devices, batteries, and sensors, [16,17,18,19] owing to its unique properties, such as high conductivity, reversibility between oxidation and reduction states, adaptability to redox states adjustments by doping, and capability of color changes according to various redox states [20]. Recently, it has been used as a material for nanocomposites capable of simultaneously measuring magnetic resonance signals and redox activity in cancer cell lines and animal models [21]. Additionally, PAni nanoparticles have been synthesized using hyaluronic acid as a surfactant. In that study, targeting the CD44 receptors expressed in cancer cell membranes and measuring the redox activity in cancer cells have been made possible [22]. The aforementioned studies used PAni nanoparticles as colorimetric nanoprobes to measure the redox activity in cancer cells through color changes according to the surrounding environment.

In this study, a lactate sensing platform was developed to sense lactate in human sweat using PAni nanoparticles and a CIELAB color system-based colorimeter to improve the detection time and portability of current gold standard methods for lactate sensing (Figure 1).

Water-soluble Tween 80-coated PAni (TPAni) nanoparticles were synthesized using the solvent shift method [23] and then physically adsorbed on the filter paper surface using the dipping method. We confirmed the colorimetric properties of TPAni nanoparticles with different pH levels bsy measuring absorbance spectra and ratios. We then use this nanoparticle to design and demonstrate a protocol for assembling TPAni and paper-based colorimetric sensors, thereby achieving short detection times and sensor portability. The availability and capability of the TPAni-based paper sensor platform were confirmed with pH tests. Finally, the color change in the filter paper sensor platform containing TPAni nanoparticles was then confirmed according to the lactate concentration and color change in the sensor platform even under the conditions in which lactate was presented in human sweat. The color change was quantified using a colorimeter that we fabricated.

## 2. Materials and Methods

### 2.1. Materials

Polyaniline (PAni, MW~5000), lactate, lactate oxidase (LOx), ferrocene, N-methyl-2-pyrrolidone (NMP), ethyl alcohol (EtOH), and Tween 80 were purchased from Sigma-Aldrich (St. Louis, MO, USA). Filter paper (grade 5c) and phosphate-buffered saline (PBS) were obtained from Advantec (Tokyo, Japan) and Welgene (Gyeongsan, Korea), respectively. All the chemicals and reagents used were of analytical grade. Ultrapure deionized water (DW) was used for all the synthetic processes.

### 2.2. Synthesis of Tween 80-Coated Polyaniline (TPAni) Nanoparticles

To synthesize TPAni nanoparticles, 5 mg PAni was first dissolved in 4 mL NMP. The resulting solution was added to 30 mL DW containing 10 mg Tween 80. The mixture was then vigorously stirred at 25 °C for 3 h. After three hours, TPAni nanoparticles were dialyzed (molecular weight cut-off: 3.5 kDa) for 48 h to remove impurities and filtered three times by centrifugation using Centricon filters (molecular weight cut-off: 3 kDa) for 30 min at 6000 rpm. The size of TPAni nanoparticles was confirmed by atomic force microscopic (AFM) imaging (NX10, Parks Systems, Suwon, Korea). The absorbance of prepared TPAni nanoparticles was measured using a UV-Vis spectrophotometer (LAMBDA 45, PerkinElmer, Waltham, MA, USA). 

### 2.3. Fabrication of TPAni Nanoparticles Containing Filter Paper Sensor Platform

The sensor platform in which TPAni nanoparticles were adsorbed on the filter paper surface was fabricated using Advantec 5c grade filter paper. The filter paper was cut (6 mm in diameter) using a punch and then immersed in an aqueous solution containing TPAni nanoparticles for 15 s before drying for 15 min at 60 °C. Subsequently, 10 µL of ferrocene dissolved in 4.5 mg/mL EtOH was dropped onto the TPAni paper sensor platform and dried for 20 min at room temperature. After drying, 10 µL LOx dissolved in 9.3 mg/mL PBS was dropped onto the TPAni paper sensor platform. 

### 2.4. Fabrication of Colorimeter

The colorimeter used in this study was fabricated using a spectrophotometer and light-emitting diode (LED) source. The spectrophotometer consists of an integrating sphere, waveguide for the coupling beam path, and spectrometer. A composite light source consisting of eight LEDs is located inside the integrating sphere. The light emitted from each LED was irradiated onto the integrating sphere through an incident light path. The light was designed to enter the integrating sphere. The coupling light path was configured to couple lights in the measurement area so that light could enter the inlet and remove stray light from the inner wall. After light was expressed from the observation area of the integrating sphere, it entered the incident optical path divided by the coupling optical path, and the incident light was divided by the spectral path and irradiated to the linear array sensor, while the other pixels of the linear distribution sensor correspond to the radiation intensity of light of different wavelengths.

### 2.5. Quantification of Color Change in TPAni Paper Sensor Platform Using Colorimeter

Ten microliters of the samples (pH solutions, lactate standard solutions, and lactate in human sweat) to be measured were treated onto the prepared TPAni paper sensor platform and then dried. The color change could be observed with the naked eye within 30 s. After drying sufficiently, the color of the TPAni paper sensor platform was quantified using a colorimeter, as mentioned in Section 2.4.

### 2.6. Human Sweat Sampling

Human sweat samples were collected from a volunteer (male, 28-year-old) and filtrated by centrifugation for 30 min at 13,500 rpm using a centricon filter (molecular weight cut-off: 10,000). After that, the filtered sweat samples were spiked with various lactate concentrations and subsequently drop-casted on a paper sensor for measurement. The remaining sweat samples were stored in the fridge before use.

## 3. Results and Discussion

### 3.1. Colorimetric Properties of TPAni Nanoparticles 

First of all, we confirmed the solubility of bare PAni in various solvents. In the case of deionized water, the bare PAni precipitated in the solution, and in the case of chloroform, the bare PAni nanoparticles had a larger size and poorer solubility than in NMP (Appendix A). Furthermore, the solubilities of TPAni nanoparticles were higher than those of bare PAni in deionized water, chloroform, and NMP (Appendix A).

Immediately after synthesizing TPAni nanoparticles, the possibility of using colorimetric probes for the TPAni nanoparticles was confirmed. As shown in Figure 2a, the TPAni nanoparticles are well-dispersed in DW without any aggregation or precipitation. The size of the TPAni nanoparticles was confirmed via an atomic force microscopic (AFM) image, and the size was 42.1 ± 5.98 nm (*n* = 100) in the AFM image (Appendix A). In addition, the stability of the TPAni nanoparticles was demonstrated using the Fourier transform infrared (FT-IR), surface-enhanced Raman scattering (SERS), and X-ray photoelectron spectroscopic (XPS) analysis, which shows TPAni nanoparticles were confirmed to be stable and conserve their chemical composition after TPAni nanoparticle synthesis [24]. 

It was confirmed that the color of the solution containing TPAni nanoparticles changed as the pH value changed; it is green under strongly acidic conditions, such as pH 1 or 2, turquoise at pH 3, and blue at pH 4 or higher. In general, PAni is green under strongly acidic conditions. The chemical state of PAni is called the emeraldine salt (ES) state, and the emeraldine base (EB) state when the pH increases. Additionally, it is known that PAni in the EB state is blue [25]. 

Furthermore, the color changes in solutions containing TPAni nanoparticles were analyzed using absorbance spectra. As displayed in Figure 2b, the absorbance spectra are nearly identical, with plateaus in the wavelength band of approximately 800 nm or higher at pH 1 or 2. At pH 3, a broad peak is observed near 800 nm; additionally, the absorbance in the visible region gradually increased. At pH 4, TPAni nanoparticles-dispersed solution is blue (Figure 2a). It is also observed in Figure 2b that the TPAni nanoparticles exist at the boundary of EB and ES states at pH 4. At pH 5 or higher, that absorbance in the near-infrared region dramatically decreases and a peak appears in the visible region (600 nm). Furthermore, the peaks at 900 and 600 nm were selected to represent the ES and EB states of PAni, respectively, for the quantification of the color change in solutions containing dispersed TPAni nanoparticles by calculating the ratio of the absorbance values at the two wavelengths. As exhibited in Figure 2c, the absorbance ratio (A_900_/A_600_) gradually decreases from pH 1 to 3, starts to rapidly decrease at pH 4, and then gradually decreases again at pH 5 and higher. This suggests that PAni is green and in the ES state when the absorbance ratio is 2.0 or higher. When the ratio is 0.25 to 2.0, PAni is turquoise as it exists in an intermediate state, and then turns blue at 0.25 or lower as it is in the EB state. 

### 3.2. Availability of PAni as a Filter Paper-Based Sensor Platform

Additional experiments were performed using PAni as a colorimetric sensor on a substrate that is not in a liquid phase, such as filter paper. We focused on ferrocene, a widely used material in electrochemical sensors that acts as a mediator to facilitate electron transfer [26]. When ferrocene is present, electron transfer becomes easy even if it is not in the liquid phase; hence, the detection time of the sensor can be drastically reduced. However, since ferrocene is orange intrinsically, we checked whether it affects the optical properties of the TPAni nanoparticles by looking into the effect of the presence and absence of ferrocene on the absorbance spectra. Ferrocene was added to the TPAni nanoparticle solution at a concentration with a minor effect on the absorbance of the TPAni nanoparticles. As shown in Figure 3a, the absorbance spectrum of the original TPAni nanoparticles (without ferrocene) is nearly similar to that in the EB state. However, although the absorbance spectrum slightly changes after ferrocene addition, it can be seen that the peak positions are similar. The absorbance ratio for each condition was calculated (Figure 3b). The absorbance ratio was 0.14 without ferrocene and 0.21 with ferrocene, which were both lower than 0.25. This implies that the PAni nanoparticles are in the EB state. Furthermore, as only a slight color difference in the solutions in these two cases is observed, the injection of ferrocene is considered to have a minor effect on the absorbance characteristics of the TPAni nanoparticles.

After ferrocene was injected, experiments were conducted to determine the absorbance properties at each pH. As shown in Figure 3c, the color of the TPAni nanoparticles changes even when ferrocene was injected, which is slightly different from the result shown in Figure 2a. As previously observed, the solution is green when the TPAni nanoparticles are in the ES state. After ferrocene injection, it was further demonstrated that it is green up to pH 4. In the conditions that appear blue, which represents PAni is EB state, the color is similar to that of Figure 2a. There is also a difference in the absorbance spectra (Figure 3d); in particular, the overall absorbance value is lower than that in Figure 2b. At pH 1 to 3 and 4 to 6, the TPAni nanoparticles were confirmed in ES and intermediate states, respectively. From pH 7, it appears to be in the intact EB state; the absorbance at the peak was 1.5 or higher in Figure 2b but approximately 1.0 in Figure 3d.

Moreover, the absorbance ratio at each pH after ferrocene injection was calculated (Figure 3e). It is observed that the absorbance ratio for the ES state is 2.0 after ferrocene injection, which is slightly lower than the 2.5 obtained in Figure 2c. However, although the absorbance properties are slightly affected, it was confirmed that the chemical states of TPAni change even after ferrocene injection.

### 3.3. Capability of TPAni Paper Sensor Platform for Colorimetric Sensing

The pH solutions were dropped on the TPAni paper sensor platform to observe color change. As shown in Figure 4a, the sensor turns turquoise at pH 1 to 4, and blue at pH higher than 4. To quantify these changes, a CIELAB color system-based colorimeter was fabricated; the colors on each substrate were measured as L*, a*, and b* (Figure 4b).

The main advantage of the CIELAB color system is its media independence, unlike RGB or CMYK. Unlike color systems, where colors vary depending on the display equipment or print media, L*a*b* color systems are defined based on research on human vision. In particular, the L* value (luminance axis), is designed to correspond to the brightness felt by humans. If L* is 0 and 100, it is black and white, respectively. a* indicates whether red or green is biased. If a* is negative and positive, it is biased toward green and red/purple, respectively. b* represents yellow and blue. If b* is negative and positive, it is blue and yellow, respectively. In addition, depending on the research results showing that human color perception is nonlinear, the L*a*b* color space has a nonlinear relationship with the actual wavelength of light. Furthermore, the distances between two different colors in the L*a*b* space are designed to be proportional to the differences in colors felt by humans.

In this study, the a* and b* values were measured and compared to focus on changes in the color itself rather than brightness. In Figure 4b, it can be seen that the * value changes with pH from approximately −10 to +2, which means that the color of the green component was strongly detected in the substrate under low pH conditions. However, the b* value varies from approximately +2 to –15, which means that the color of the blue component appears stronger as the pH value increases. However, it was confirmed that TPAni nanoparticles were physically adsorbed on the filter paper as the color remarkably changed without ferrocene (Figure 4c). Blue is observed at pH 5 or higher, which is similar to that shown in Figure 4a. However, at pH 1 to 4, it is green. In the color quantification experiment for each substrate, using the colorimeter shown in Figure 4d, the difference is comparable to that shown in Figure 4b. The a* and b* values change from approximately –12 to +20 and –20 to 0, respectively, according to pH. Notably, a* value was detected more positively under high pH. This can be attributed to the ferrocene injection causing more changes in the red/purple component. The subsequent experiments were conducted using ferrocene. Results confirmed that TPAni nanoparticles could be used as colorimetric sensing probes, even on filter paper. 

### 3.4. Colorimetric Ability of TPAni Paper Sensor Platform for Sensing Lactate in Human Sweat

To verify the lactate sensing capability of the TPAni paper sensor platform, ferrocene and LOx were sequentially injected after the TPAni nanoparticles were physically attached to the filter paper. Lactate with varying concentrations in the range of 1 to 100 mM was dropped on the TPAni paper sensor platform containing ferrocene and LOx (Figure 5a). At low (1 mM or lower) and high lactate concentrations (10 and 100 mM), bright green and dark blue were observed, respectively. Color quantification was also performed using a colorimeter (Figure 5b). The a* and b* values ranged from approximately –12 to 0 and –15 to +20, respectively. Compared with previous experiments using pH solutions, both a* and b* values have dramatically decreased in the positive portion. This indicates that the red and yellow portions of each value were greatly reduced; only the blue and green portions affected the color change. Even if the detection limit of lactate is approximately 100 µM, this concentration still exists within a range that has an important meaning in the concentration of lactate in the human body [27]. This phenomenon can be explained by the following equations: (1)$$\mathrm{lactate}+H_{2}O+O_{2}\overset{LOx}{\to }\mathrm{pyruvate}+H_{2}O_{2}$$
(2)$$H_{2}O_{2}+\mathrm{PAni}\left(\mathrm{EB}\right)\to 2e^{-}+O_{2}+\mathrm{PAni}\left(\mathrm{ES}\right)$$

In Equation (1), lactate is oxidized by LOx, resulting in pyruvate and hydrogen peroxide. Moreover, as shown in Equation (2), the resulting hydrogen peroxide is reduced from the PAni in the EB state to the ES state. In other words, the PAni-based sensor platform changes color according to the redox state of PAni.

Furthermore, lactate was spiked into human sweat and dropped on the TPAni paper sensor platform (Figure 5c). The human sweat samples were collected from volunteers, and the sweat samples were spiked with specific lactate concentrations. Sweat was obtained in the heating chamber because exercising affects the concentration of lactate. The color of the TPAni paper sensor platform is faded as a whole; however, there is a color difference according to the lactate concentration, as in the standard solution (Figure 5a). It was confirmed that the a* and b* values range from –10 to –4 and –5 to +20, respectively. In the case of the a* value, it was confirmed that the value only existed in the negative region, whereas a large number of negative regions disappeared in the case of b*. Under these conditions, the combination of polyaniline-based sensor platforms and colorimetric systems has LOD and LOQ of 1 mM, linearity of 0.9684, and stability of 14% (Appendix A). Moreover, the color change of the polyaniline-based sensor platform proceeded in about 30 s. As explained earlier, this can be attributed to the fading of color when applied to experiments using human sweat. That is, it is assumed that various components in sweat caused the TPAni paper sensor platform to fade. Nevertheless, based on these results, it was confirmed that lactate sensing is possible using the TPAni paper sensor platform. Additionally, a clinically meaningful lactate concentration could be determined even in human sweat using a combination of the TPAni paper sensor platform and colorimeter. The lactate concentration when not exercising is several mM scales. In addition, it is known that lactate in sweat is detected on average at 43.7 mM and 115.8 mM during endurance and intense exercise, respectively [27,28]. The linearity of the combination of the PAni-based sensor platform and colorimeter has been confirmed up to 1 to 100 mM, so the aforementioned lactate concentrations can be detected sufficiently.

## 4. Conclusions

In this study, we successfully synthesized PAni nanoparticles using the solvent shift method. The color change characteristics of PAni according to the changes in the surrounding environment were confirmed. In addition, the ferrocene concentration was optimized, and the color change in the TPAni paper sensor platform was quantified using a colorimeter that we fabricated. Color changes associated with lactate concentration were also confirmed, suggesting the possibility of detecting lactate in human sweat. Moreover, it was confirmed that the detection limit for lactate was about 10 mM. Studies related to the detection of disease biomarkers with a more essential medical significance are being conducted. TPAni nanoparticles could easily be adapted to the detection of other metabolites when the corresponding redox signaling is available. Therefore, future innovations in this study will focus on the multiplex colorimetric detection of various types of metabolites involved in clinical diagnostics and environmental monitoring on one paper device by integrating paper-based microfluidics systems.

## Figures and Tables

**Figure 1 biosensors-12-00248-f001:**
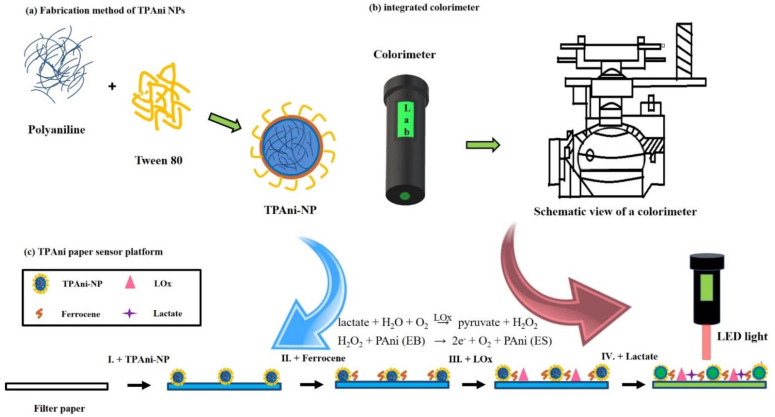
Schematic of: (**a**) synthetic process of the TPAni nanoparticles (TPAni-NPs), (**b**) description of the colorimeter, and (**c**) lactate sensing mechanism of the TPAni-NP-adsorbed paper sensor platform.

**Figure 2 biosensors-12-00248-f002:**
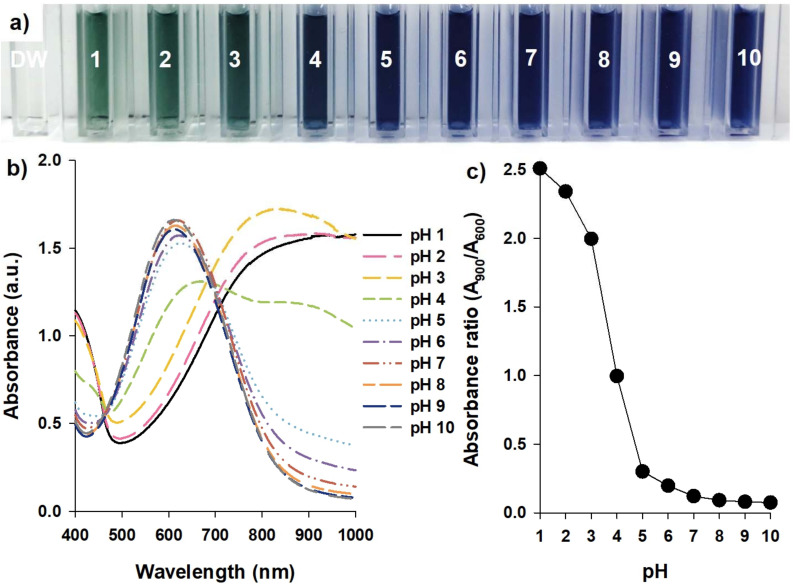
(**a**) Photograph, (**b**) absorbance spectra, and (**c**) absorbance ratio of the TPAni nanoparticles in varying surrounding pH.

**Figure 3 biosensors-12-00248-f003:**
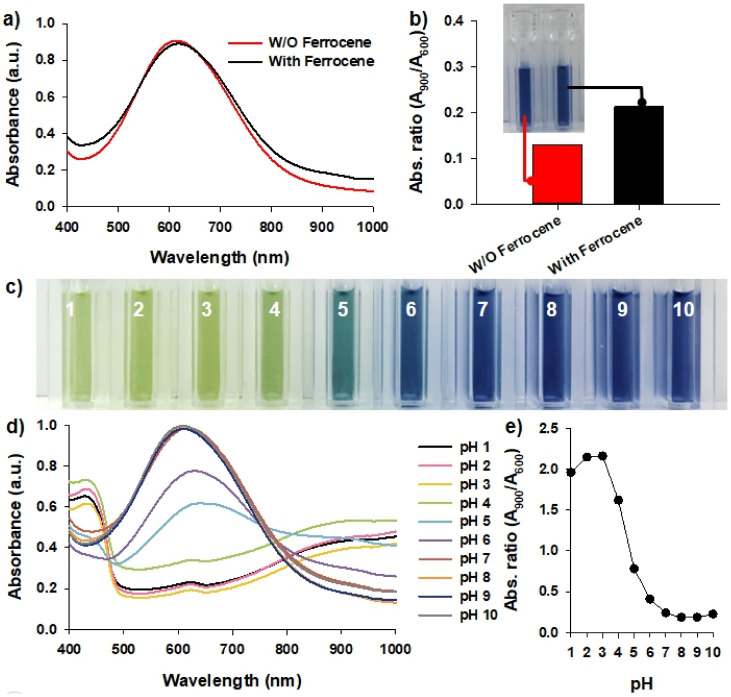
(**a**) Absorbance spectra, (**b**) absorbance ratios (A_900_/A_600_) about TPAni nanoparticles with or without ferrocene (Inset: photograph of TPAni solutions), (**c**) photograph, (**d**) absorbance spectra, and (**e**) absorbance ratios (A_900_/A_600_) of TPAni nanoparticles with ferrocene in pH solutions (from pH 1 to 10).

**Figure 4 biosensors-12-00248-f004:**
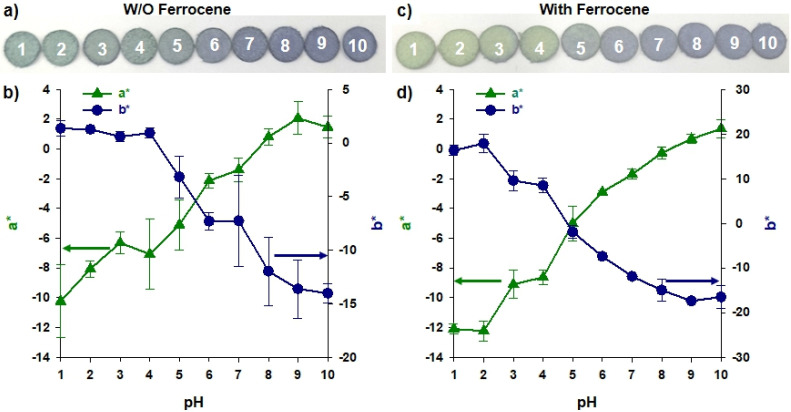
(**a**) Photograph and (**b**) a* and b* values of each TPAni paper sensor platform according to pH (*n* = 10). (**c**) A photograph and (**d**) a* and b* values of each TPAni paper sensor platform according to pH after ferrocene treatment (*n* = 10).

**Figure 5 biosensors-12-00248-f005:**
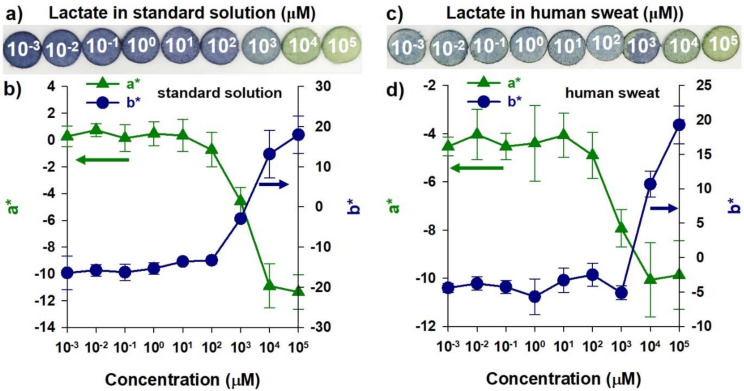
(**a**) Photograph and (**b**) a* and b* values of each TPAni paper sensor platform according to lactate concentration using standard solutions (*n* = 10). (**c**) A photograph and (**d**) a* and b* values of each TPAni paper sensor platform using various lactate concentrations in human sweat (*n* = 10).

## Data Availability

Data sharing is not applicable.

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
