# Peer review of "Colorimetric Sensing of Lactate in Human Sweat Using Polyaniline Nanoparticles-Based Sensor Platform and Colorimeter"

_biosensors, 2022, doi:10.3390/bios12040248_

Round 1

Reviewer 1 Report

The paper entitled ‘Colorimetric sensing of lactate in human sweat using polyaniline nanoparticles-based sensor platform and colorimeter’ reported a polyaniline (PAni)-based single-sensor platform for sensing lactate in human sweat using a CIELAB color system-based colorimetric device. To establish a lactate sensing platform, PAni nanoparticles were synthesized and adsorbed on the filter paper surface using solvent shift and dip-coating methods, respectively. To quantify the color change of PAni, a CIELAB color system-based colorimetric device was fabricated. The color change of PAni was measured according to the chemical state using a combination of a PAni-based filter paper sensor platform and colorimetric device, based on the lactate concentration in deionized water. Finally, human sweat was spiked with lactate to measure the color change of the PAni-based filter paper sensor platform. This is an interesting work. I can recommend its publication after the following questions are addressed.

  1. In the abstract the LOD, LOQ, linearity and the stability should be described.
  2. The author should clarify what caused the color change. Is it the redox state of the PAni on the paper or the pH that affects the color of PAni? How does the redox reaction of lactate influence the redox state or pH?
  3. Please show the particle size with the image by SEM or TEM.
  4. How is the water solubility of this PAni? Can it be directly dissolved in water?
  5. How do you get the sweat samples for the spiking experiment?
  6. There is no specificity experiment. Please add this section.
  7. The sensor performance is not mentioned in terms of its linearity, Time of response, LOD and LOQ. Please add these sections.
  8. Please specify whether the sensor's detection range can meet the requirement for the detection of lactate for clinical use. List the linear range and the typical clinical concentration of lactate acid.

Author Response

  1. In the abstract the LOD, LOQ, linearity and the stability should be described.

A: We appreciate to reviewer’s indications. Among the parameters mentioned by the reviewer, first, the LOD and LOQ are 1 mM. Also, the value of R2, which represents linearity, is 0.9684, and the stability of the sensor exists within 14% of the average under the conditions of the largest error range.

We added the above to abstract and Figure S3 to explain the parameters mentioned by reviewers for better understanding of readers. Of course, the content of the text was also added.

Inserted sentences (Page 1, Line 29)

Under these conditions, the combination of polyaniline-based sensor platforms and colorimetric systems has a limit of detection (LOD) and limit of quantitation (LOQ) of 1 mM, linearity of 0.9684, and stability of 14%. Moreover, it can be confirmed that the color of the substrate changes after about 30 s, and through this, the physical fatigue of the individual can be determined.

Inserted sentences (Page 9, Line 286)

Under these conditions, the combination of polyaniline-based sensor platforms and col-orimetric systems has LOD and LOQ of 1 mM, linearity of 0.9684, and stability of 14% (Figure S3). Moreover, the color change of the polyaniline-based sensor platform proceed-ed in about 30 seconds.

Added Figure S3

  1. The author should clarify what caused the color change. Is it the redox state of the PAni on the paper or the pH that affects the color of PAni? How does the redox reaction of lactate influence the redox state or pH?

A: We thank to reviewer’s comments. To help reviewers and readers understand, the chemical reaction equations for LOx and lactate used in the experiment were added to Figure 1, and explanatory statements were also inserted.

Modified Figure 1 (Page 3, Line 87)

Inserted sentences (Page 8, Line 271)

(1)

(2)

This phenomenon can be explained by the following equations:

In equation (1), lactate is oxidized by LOx, resulting in pyruvate and hydrogen peroxide. Moreover, as shown in equation (2), the resulting hydrogen peroxide is reduced the PAni in EB state to ES state. In other words, the PAni-based sensor platform changes color according to the redox state of PAni.

  1. Please show the particle size with the image by SEM or TEM.

A: We appreciate to reviewer’s comment. The size of the TPAni nanoparticles was confirmed using AFM rather than the imaging device mentioned by the reviewer, and this image was added as Figure S2.

Added Figure S2

Inserted sentences (Page 4, Line 150)

The size of the TPAni nanoparticles was confirmed via atomic force microscopic (AFM) image, and the size was 42.1 ± 5.98 nm (n = 100) in the AFM image (Figure S2).

  1. How is the water solubility of this PAni? Can it be directly dissolved in water?

A: We thank to reviewer’s comments. It was confirmed that polyaniline had high solubility in the order of NMP, chloroform, and water, and this result was added as Figure S1.

Added Figure S1

Inserted sentences (Page 4, Line 142)

First of all, we confirmed the solubility of bare PAni in various solvents. In the case of deionized water, the bare PAni precipitated in the solution, and in the case of chloroform, the bare PAni nanoparticles had larger size and poorer solubility than in NMP (Figure S1a. Furthermore, the solubilities of TPAni nanoparticles were higher those of bare PAni in deionized water, chloroform, and NMP (Figure S1b and c).

  1. How do you get the sweat samples for the spiking experiment?

A: We appreciate to reviewer’s comment. The process of obtaining human sweat samples is added to the main text in manuscript.

Inserted sentences (Page 8, Line 279)

The human sweat samples were collected from volunteers, and the sweat samples were spiked with specific lactate concentrations. Sweat was obtained in the heating chamber because exercising affects the concentration of lactate.

  1. There is no specificity experiment. Please add this section.

A: We thank to reviewer’s comments. The process of LOx oxidizing lactate into pyruvate is a well-known fact. In addition, numerous papers related to electrochemical-based biosensors that used this reaction to detect lactate have also been published [(1993) Biosens. Bioelectron. 8 409, (1997) Biosens. Bioelectron. 12 539, (2002) Sens. Actuat. B 82 227, (2009) Anal. Biochem. 384 159, (2006) Sens. Actuat. B 124 269]. Looking at these papers, it can be seen that LOx has a very strong selectivity for lactate. Therefore, we think it is not necessary to show the specificity requested by the reviewer.

  1. The sensor performance is not mentioned in terms of its linearity, Time of response, LOD and LOQ. Please add these sections.

A: We appreciate to reviewer’s comments. It is a question similar to the first question you asked, and the parameters are presented in the main text.

Inserted sentences (Page 9, Line 286)

Under these conditions, the combination of polyaniline-based sensor platforms and colorimetric systems has LOD and LOQ of 1 mM, linearity of 0.9684, and stability of 14% (Figure S3). Moreover, the color change of the polyaniline-based sensor platform proceeded in about 30 seconds.

  1. Please specify whether the sensor's detection range can meet the requirement for the detection of lactate for clinical use. List the linear range and the typical clinical concentration of lactate acid.

A: We thank to reviewer’s valuable comments. In other research papers [(2012) J. Physiol. Sci. 62 429, (2010) Bull. Exp. Biol. Med. 150 83], the lactate concentration when not exercising is several mM scale. In addition, it is known that lactate in sweat is detected on average 43.7 mM and 115.8 mM during endurance and intense exercise, respectively. The linearity of combination of PAni-based sensor platform and colorimeter has been confirmed up to 1 to 100 mM, so the aforementioned lactate concentrations can be detected sufficiently.

We inserted this content into the main text.

Inserted sentences (Page 9, Line 296)

The lactate concentration when not exercising is several mM scale. In addition, it is known that lactate in sweat is detected on average at 43.7 mM and 115.8 mM during endurance and intense exercise, respectively [25, 26]. The linearity of combination of PAni-based sensor platform and colorimeter has been confirmed up to 1 to 100 mM, so the aforementioned lactate concentrations can be detected sufficiently.

Reviewer 2 Report

Overall this is an interesting paper with well designed experiments, and interesting results.  I do think the paper could use a few edits and changes to improve its readability, as well as to strengthen the arguments and use case of the proposed paper sensor.  I have prepared the following suggestions and questions I think the paper should answer to strengthen it overall.

  • Minor spelling and grammar issues throughout, needs a quick language edit.
  • Final paragraph of the introduction should state the hypothesis, and proposed experiments, but not include results as it does now.
  • Materials section needs locations of the companies listed so we know which one each was purchased from.
  • Section 2.2 – was the cutoff 3 kDa for both dialysis and for centrifugation?
  • Section 2.4 – would be useful to include a diagram of the developed colorimeter as it would make it clearer how it was constructed
  • Section 3.1 – would be nice to split this section into a few paragraphs, as it is somewhat hard to read as a single block of text
  • Figure 2b, is a little difficult to differentiate lines, line weights or dashing should be varied to make it clearer which line corresponds to which pH.
  • Section 3.3 – same note as 3.1, multiple paragraphs should be used to improve readability.
  • Figure 5a and c – What units are the concentrations in?
  • Section 3.4 – what were the exact limits of detection calculated for each condition?
  • What is the biologically relevant range of lactate concentration, and how does your LoD compare to that range?
  • For each section and experiment the number of replicates must be stated.
  • Conclusions should include some more discussion of future work, and specific ideas and experiments that will continue from the presented work.

Author Response

Comments and Suggestions for Authors

Overall this is an interesting paper with well designed experiments, and interesting results.  I do think the paper could use a few edits and changes to improve its readability, as well as to strengthen the arguments and use case of the proposed paper sensor.  I have prepared the following suggestions and questions I think the paper should answer to strengthen it overall.

  • Minor spelling and grammar issues throughout, needs a quick language edit. 
  • A: Thanks for your comment. We double-checked the spelling and grammar in the manuscript.
  • Final paragraph of the introduction should state the hypothesis, and proposed experiments, but not include results as it does now. In this study, a lactate sensing platform was developed to sense lactate in human sweat using PAni nanoparticles and a CIELAB color system-based colorimeter to improve the detection time and portability of current gold standard methods for lactate sensing (Figure 1). 
  • Water-soluble Tween 80-coated PAni (TPAni) nanoparticles were synthesized using the solvent shift method [22] and then physically adsorbed on the filter paper surface using the dipping method. We confirmed the colorimetric properties of TPAni nanoparticles with different pH levels by measuring absorbance spectra and ratios. We then use this nanoparticle to design and demonstrate a protocol for assembling TPAni and paper-based colorimetric sensors, thereby achieving short detection time and sensor portability. The availability and capability of the TPAni-based paper sensor platform were confirmed with pH tests. Finally, the color change in the filter paper sensor platform containing TPAni nanoparticles was then confirmed according to the lactate concentration and color change in the sensor platform even under the conditions in which lactate was presented in human sweat. The color change was quantified using a colorimeter that we fabricated.
  • Modified sentences (Page 2, Line 65)
  • A: A: We appreciate to the reviewer’s comment. The final paragraph of the introduction was modified as the reviewer suggested.
  • Materials section needs locations of the companies listed so we know which one each was purchased from. Polyaniline (PAni, MW~5,000), lactate, lactate oxidase (LOx), ferrocene, N-methyl-2-pyrrolidone (NMP), ethyl alcohol (EtOH), and Tween 80 were purchased from Sigma-Aldrich (St. Louis, USA). Filter paper (grade 5c) and phosphate-buffered saline (PBS) were obtained from Advantec (Tokyo, Japan) and Welgene (Gyeongsan, Korea), respectively.
  •  
  • Modified sentences (Page 3, Line 86)
  • A: We appreciate to reviewer’s comments. We added the manufacturer's location to the Materials section.
  • Section 2.2 – was the cutoff 3 kDa for both dialysis and for centrifugation? After three hours, TPAni nanoparticles were dialyzed (molecular weight cut-off: 3.5 kDa) for 48 h to remove impurities and filtered three times by centrifugation using Centricon filters (molecular weight cut-off: 3 kDa) for 30 min at 6,000 rpm.
  •  
  • Modified sentences (Page 3, Line 96)
  • A: We thank to reviewer’s comments. It was confirmed that molecular weight cut-off of the dialysis membrane was omitted during the manuscript editing process. The molecular weight cut-off for this dialysis membrane was added.
  • Section 2.4 – would be useful to include a diagram of the developed colorimeter as it would make it clearer how it was constructed 
  • A: We appreciate to reviewer’s comment. As you can see, there is a schematic diagram of the colorimeter in Figure 1. For the purpose of understanding of reviewers and readers, we think it is right to show the diagram of the colorimeter we have developed. However, please understand that it is difficult to show any further details because it will cause problems with the patents we are currently carrying on.
  • Section 3.1 – would be nice to split this section into a few paragraphs, as it is somewhat hard to read as a single block of text. Split paragraph (Page 4, Line 134, 143, and 149)
  •  
  • A: A: Thanks for your valuable suggestion. Section 3.1 was split into four paragraphs.
  • Figure 2b, is a little difficult to differentiate lines, line weights or dashing should be varied to make it clearer which line corresponds to which pH.  
  • Modified Figure 2 (Page 5, Line 171)
  • A: We thank to reviewer’s comments. As suggested by the reviewer, the difference in the absorbance spectrum for each pH was made by changing the dash type for the absorbance spectra in Figure 2b.
  • Section 3.3 – same note as 3.1, multiple paragraphs should be used to improve readability.Split paragraph (Page 7, Line 213 and Line 224)
  •  
  • A: Thanks for your valuable suggestion. Section 3.1 was split into three paragraphs.
  • Figure 5a and c – What units are the concentrations in?  
  • Modified Figure 5 (Page 9, Line 294)
  • A: We appreciate to reviewer’s indication. For a clear understanding of Figure 5a and c, the concentration of lactate was added.
  • Section 3.4 – what were the exact limits of detection calculated for each condition?  
  • Added Figure S3
  • A: We thank to reviewer’s comment. The b*/a* value was calculated for Figure 5c and d, which can be confirmed for the possibility of detecting lactate in substantial human sweat, and it was added as Figure S3. The detection limit in this case was 1 mM.
  • What is the biologically relevant range of lactate concentration, and how does your LoD compare to that range?We inserted this content into the main text.Inserted sentences (Page 9, Line 289) 
  • The lactate concentration when not exercising is several mM scale. In addition, it is known that lactate in sweat is detected on average at 43.7 mM and 115.8 mM during endurance and intense exercise, respectively [25,26]. The linearity of combination of PAni-based sen-sor platform and colorimeter has been confirmed up to 1 to 100 mM, so the aforemen-tioned lactate concentrations can be detected sufficiently.
  •  
  • A: We appreciate to reviewer’s comments. In other research papers [(2012) J. Physiol. Sci. 62 429, (2010) Bull. Exp. Biol. Med. 150 83], the lactate concentration when not exercising is several mM scale. In addition, it is known that lactate in sweat is detected on average 43.7 mM and 115.8 mM during endurance and intense exercise, respectively. The linearity of combination of PAni-based sensor platform and colorimeter has been confirmed up to 1 to 100 mM, so the aforementioned lactate concentrations can be detected sufficiently.
  • For each section and experiment the number of replicates must be stated. Figure 4. a) Photograph and b) a* and b* values of each TPAni paper sensor platform according to pH (n=10). c) A photograph and d) a* and b* values of each TPAni paper sensor platform ac-cording to pH after ferrocene treatment (n=10).Modified Figure 5 caption (Page 9, Line 295) 
  • Figure 5. a) Photograph and b) a* and b* values of each TPAni paper sensor platform according to lactate concentration using standard solutions (n=10). c) A photograph and d) a* and b* values of each TPAni paper sensor platform using various lactate concentrations in human sweat (n=10).
  •  
  • Modified Figure 4 caption (Page 8, Line 248)
  • A: We thank to reviewer’s indications. In the case of a repetitive experiment, the number of repetitions was written in each figure caption.
  • Conclusions should include some more discussion of future work, and specific ideas and experiments that will continue from the presented work. TPAni nanoparticles could easily be adapted to the detection of other metabolites when the corresponding redox signaling is available. Therefore, future innovations in this study will focus on the multiplex colorimetric detection of various types of metabolites involved in clinical diagnostics and environmental monitoring on one paper device by integrating paper-based microfluidics systems.
  • Modified Figure 5 caption (Page 10, Line 307)
  • A: We really appreciate your comment. We added the applicability and future direction of our work in the conclusion section

Reviewer 3 Report

As the attachment.

Author Response

  1. The Abstract said “Clinical lactate measurements generally require 3 h for clinical determination”, the authors should address how long this measurement would take when using the proposed device.

A: We appreciate to reviewer’s comments. Using a combination of a PAni-based sensor platform and a colorimeter, it can be seen that the color of the substrate changes about 30 seconds after the sample is administered.

This content was added to the abstract.

Added sentences (Page 1, Line 31)

Moreover, it can be confirmed that color of the substrate changes after about 30 s, and through this, the physical fatigue of the individual can be determined.

  1. The Abstract section is ambiguous about the purpose of the experiment, and could be improved to more concise.

A: We appreciate your comment. As the reviewer suggested, the main purpose of our study is the development of TPAni-based colorimetric sensors with having short detection time and good portability. The above description was added in the abstract.

Added sentences (Page 1, Line 18)

To improve the current gold standard methods, the development of sensor devices that can reduce detection time while maintaining sensitivity and providing portability is gaining great attention.

  1. The instruments used in this manuscript should be listed in experimental section.

A: We thank to reviewer’s comments. After TPAni nanoparticles were synthesized, the size was measured using atomic force microscope, and absorbance was also measured. The instruments used at this time were added to the main text.

Added sentences (Page 3, Line 98)

The size of TPAni nanoparticles was confirmed by atomic force microscopic (AFM) imaging (NX10, Parks Systems, Korea). The absorbance of prepared TPAni nanoparti-cles was measured using a UV-Vis spectrophotometer (LAMBDA 45, Perkin Elmer, USA).

  1. The structures of tween 80-coated polyaniline (TPAni) nanoparticles and TPAni nanoparticles synthesized should be characterized in order to guarantee the stability of these nanoparticles, which is very important for clinical measurements in emergency medicine.

A: Thanks for your suggestion. Our group recently demonstrated the component of the TPAni nanoparticles using the Fourier transform infrared (FT-IR), surface-enhanced Raman scattering (SERS), and X-ray photoelectron spectroscopic (XPS) analysis. Predominant peaks of PAni, tween 80, and their bonding were detected, which means TPAni nanoparticles were confirmed to be stable and conserve their chemical composition after TPAni nanoparticle synthesis.

Added sentences (Page 4, Line 143)

In addition, the stability of the TPAni nanoparticles were demonstrated using the Fourier transform infrared (FT-IR), surface-enhancd Raman scattering (SERS), and X-ray photoelectron spectroscopic (XPS) analysis, which shows TPAni nanoparticles were confirmed to be stable and conserve their chemical composition after TPAni nanoparticle synthesis [23].

Round 2

Reviewer 1 Report

As my concerns have been responded. I believe it can be published in the current version. 

Author Response

We are pleased with the reviewer's response.